# The associations of palliative care experts regarding food refusal: A cross-sectional study with an open question evaluated by triangulation analysis

**André Fringer**[1,2]*, **Sabrina Stängle**[1,2], **Daniel Büche**[3], **Stefan Ch. Ott**[4], **Wilfried Schnepp**[2†]

1 Institute of Nursing, School of Health Professionals, ZHAW University of Applied Sciences, Winterthur, Switzerland, 2 Department of Nursing Science, Institute of Health, Witten/Herdecke University, Witten, Germany, 3 Palliative Centre St.Gallen, Cantonal Hospital St.Gallen, St.Gallen, Switzerland, 4 Department of Economics, FHS St.Gallen University of Applied Sciences St.Gallen, St.Gallen, Switzerland

† Deceased.
* andre.fringer@zhaw.ch

**Data Availability Statement:** The written answers of the participants may not be passed on on the basis of the submissions of the Ethics Committee (Ethikkommission Ostschweiz - EKOS, office:

## Abstract

### Introduction

Health professionals in oncologic and palliative care settings are often faced with the problem that patients stop eating and drinking. While the causes of food refusal are very different, the result is often malnutrition, which is linked to health comorbidities and a high mortality rate. However, the professionals lack the time and knowledge to clarify the cause for each patient. What associations do health professionals have when faced with food refusal?

### Objective

To investigate the associations that health professionals in oncological and palliative settings have about denied eating behavior

### Methods

A cross-sectional study, starting with an open question focusing professionals' associations regarding food refusal. The results were inductively analyzed, whereby generic categories were developed. Subsequently, the categories were transformed into quantitative data to calculate the relationships between the categories.

### Results

A total of 350 out of 2000 participants completed the survey, resulting in a response rate of 17.5%. *Food refusal* is primarily associated with physical and ethical aspects and with end-of-life. Half of the participants frequently find that patients refuse to eat. The attitudes show that the autonomy of the patient is the highest good and is to be respected. Even in the case of patients with limited decision-making capacity, the refusal to eat is acceptable.

sekretariat@ekos.ch). All information required for the calculation of the data is contained in the manuscript.The data set represents a large target group in the relatively small country of Switzerland. Accordingly, the conclusions of the present study have a high significance for Switzerland. In order to repeat the study, we specified the exact question, narrowed down the setting and sample and described the study participants in great detail, including mean values and standard deviations. The analysis starts with an inductive content analysis, the categorization of which is shown in Table 2. For the further steps, we have attached great importance to describing the data analysis very precisely, as it is partly an as yet unknown analysis method. The individual steps are comprehensibly structured by tables and explanatory words and are documented by absolute and relative frequencies and p-values. The validation of the findings is done in the last step, with the substantiation of the findings by the qualitative statements of the participants.

**Funding:** A.F. reports grant vom Hans Altschüler-Foundation, Switzerland. https://stiftungschweiz.ch/organisation/dr-hans-altschueler-stiftung The funders had no role in study design, data collection and analysis, decision to publish, or preparation of the manuscript.

**Competing interests:** A.F. reports grant vom Hans Altschüler-Foundation, Switzerland. The authors A. F., S.S., D.B., O.C.S. and W.S. have nothing to disclose.

**Abbreviations:** SENS model, SENS is a German abbreviation and stands for Symptom-Management (symptom management), Entscheidungsfindung (decision making), Netzwerk (networking) and Support (support).

## Conclusion

Clarifying the cause of food refusal requires a great deal of knowledge and is strongly influenced by the associations of health professionals. While the associations have very negative connotations, information and training is needed to make professionals aware of this and to change their associations. With this knowledge and in an interprofessional cooperation, mis-labelling of patient settings can be avoided and fears can be reduced.

## Introduction

The intake of food is a daily act in which both the physical needs are satisfied and social interaction takes place [1]. In contrast, the refusal of food and liquids is closely linked to the disease and its associated effects [2]. In this respect, it is not surprising that food refusal often occurs in clinics and long-term care, especially in the oncological and palliative setting [3–8]. For health professionals who work in these areas, dealing with people who refuse to eat is part of their daily business and poses a major challenge [9]. Professionals must therefore develop strategies for dealing with such situations. It is well known that the reasons for food refusal can be due to social, psychological and medical reasons or a combination thereof [5, 6, 9, 10]. Even the institutionalization itself can trigger food refusal through its day-to-day structure. People get used to their individual eating habits over the course of their lives. However, upon entry into an institution, food times are predetermined, and the selection of food is limited. The eating habits of the individual cannot be fully taken into account. Moreover, the patient may not like the food, may have no appetite, may be interrupted while eating or may feel uncomfortable in the location [6, 11, 12]. It should also be taken into account that, regardless of physiological changes, appetite and food intake change in old age [13]. Cancer patients can develop a food aversion due to pain and discomfort felt while eating or because of cancer treatments, such as chemotherapy and radiotherapy [8]. Furthermore, the patients may refuse food in protest against the nurses or in form of an implicit or unspoken stopping of eating and drinking with the intention of dying [5, 11, 14–17]. From the various causes, it emerges that the approach to treat people who refuse to ingest food appropriately requires multidimensional treatment [18]. Which approach is chosen depends on how the health professionals assess the situation. Therefore, it is of interest to know what considerations and associations guide the professionals to their decisions. This research question is the subject of this work.

### Review of the literature

Food refusal und the related malnutrition are directly linked to health comorbidities and a high mortality rate [19]. Given that approximately 30% of institutionalized elderly individuals [20] and 40 to 80% of oncological patients [8] suffer from malnutrition, health care professionals are often confronted with the decision to clarify the cause for why the patient refuses to eat. A lack of knowledge about nutrition and the nurses' high workload can lead to underdiagnosed and undertreated malnutrition [21–23]. The systematic use of assessment instruments, such as Nutritional Risk Screening (NRS 2002) [24]; the Short Nutritional Assessment Questionnaire (SNAQ) [25]; the Mini Nutritional Assessment (MNA) [26]; or the Malnutrition Universal Screening Tool (MUST) [27], can help to identify malnutrition in patients [28]. However, what if the situation is not recognized or is misinterpreted? We asked health professionals about their initial associations regarding how to deal with the situation when patients refuse food.

## Objectives

The aim of this study was to investigate the associations that health professionals in oncological and palliative settings have about denied eating behavior.

## Methods

### Design

We designed a web-based cross-sectional study [29], starting with an open question, followed by quantitative questions. This article limits itself to the evaluation of the open question using the unit of analysis triangulation [30], starting with a classical qualitative data analysis [31] followed by transformation of the qualitative data into numeric counts (quantitative data).

### Sample

Health professionals working in palliative and oncological settings were invited to the study by gatekeepers and through *palliative ch*, a Swiss Society for Palliative Medicine, Care and Accompaniment [32]. The society currently has over 2000 members and is aimed at all professional groups involved in the care of patients in palliative care, including physicians, nurses, pastors, and volunteers.

### Data collection

Based on a literature review in the databases PubMed and CINAHL, we developed an anonymized self-completion survey. The survey was carried out between 2015 and 2016 using the survey software QuestBack (EFS survey).

Starting with the open question, we asked professionals to react spontaneously to the following statement and to write down their associations and ideas based on their professional experiences and their contact with patients:

*"Food refusal–and the lips remain closed!"*

The participants could write their answer in a free answer field. This requires a qualitative evaluation of the data. Through the use of an open question, we wanted to portray food refusal as it is actually perceived by the participants [33].

In addition to this open question, the questionnaire contains another 119 items on stressful experiences of caring relatives and health professionals in refusing food and attitudes and experiences of health professionals regarding voluntary stopping of eating and drinking (VSED). This article presents the results of the open question.

### Institutional review board

Interviews with experts are not covered by the law on human research (Humanforschungsgesetz, HFG) in Switzerland section 2 [1] sentence 2c HFG, since no health-related data were collected. In this respect, no ethical vote could be obtained from the ethics committee for this research. The ethical approach for the present survey is based on the principles of the Declaration of Helsinki and informed consent. Anonymity and respect for human dignity were guaranteed at all times during the research process. Drawing any conclusions about the respondents was not possible at any time [34]. After a brief introduction about the necessity and the aim of this study, the participants were informed about their safety and anonymity and were provided reasons for why they should answer this questionnaire. With the continuation of the questionnaire, informed consent was assumed.

## Data analysis

Descriptive data analysis to describe the study participants was evaluated using established methods using the software IBM SPSS Statistics (version 23).

The data analysis of the open question follows the principles of the unit of analysis triangulation [30]. The data set was evaluated and reported using various analysis methods. In the present case, two methods were applied. In the first phase of the analysis, the principles of qualitative descriptive research methods [35, 36] were pursued. An inductive content analysis [37] was conducted, and generic categories were developed [38]. Then, the material was analyzed by keywords in a context analysis [39]. Finally, sub-categories and categories were developed. In the second phase, the sub-categories and categories (qualitative data) were transformed into quantitative data (total and relative frequencies) [40, 41] to gain a deeper understanding of the relationship between the categories and sub-categories [42]. The relations between the sub-categories and categories were calculated using the Code Relations Browser (CRB) within the software MAXqda11 [43]. The CRB is a visualization of the relationships between codes. The CRB calculated the immediate proximity of codes and indicated to what extent individual codes were consistent with each other. Thereby, statements concerning connections between phenomena were possible. We calculated the distance of two codes from each other in a maximum distance of one paragraph [44]. Independence between lines and columns was investigated by the chi-square test. The code relations were tested to determine whether the relations were a random effect or a real and therefore significant connection between the codes. The level of significance of the code relations was calculated using Fisher's exact test (significance level alpha = .05). After quantitizing the qualitative data, the categories and sub-categories and the conceptual model regarding the keyword associations were developed by using the significances of the code relations to answer the research question.

## Results

### Sample characteristics

In total, 350 of 2000 participants completed the questionnaire, resulting in a response rate of 17.5%. As is apparent from the socio-demographic and professional data in Table 1 below, the majority of participants (81.9%) were female. Almost three-quarters (72.8%) of the participants were between 31 and 55 years old. While one-quarter of the participants ($n = 58$; $\bar{x} = 26.1$) did not provide further information about their education in palliative care, three-quarters were educated at several competence levels. Most of the participants were nurses (72.8%), working in hospitals (63.1%) and mainly taking care of patients with cancers (32.5%) and patients in palliative care (25.5%). The average work experience was 16 years (Min = 0; Max = 45; SD = 11.867).

First, the analysis of the keyword-related associations will be reported. Furthermore, we identified three themes regarding experiences and another three themes regarding attitudes from our data.

### Associations with food refusal among health professionals

Health care professionals responded to the request to relate about food refusal with associations concerning "physical aspects", (31%), "end-of-life aspects" (26.3% ), "ethical aspects" (19%), "mental aspects" (17.9%) and "modes of reactions and behavior" (5.1%) (Table 2).

Words attributable to professionals' modes of reactions and behavior were related to *meal preferences* ($n = 2$), *lip care* ($n = 1$), *acceptance* ($n = 3$), and *assuming* ($n = 2$). In contrast, participants mentioned terms like *consequence* ($n = 2$), *endurance* ($n = 3$) and *persistence* ($n = 1$).

**Table 1. Characteristics of the participants.**

| Variable | | N | % |
|---|---|---|---|
| **Gender (*n* = 221):** | **Female** | **181** | **81.9%** |
| | **Male** | **40** | **18.1%** |
| Age (*n* = 221) | < 30 years | 27 | 12.2% |
| | 31–45 years | 88 | 39.8% |
| | 46–55 years | 73 | 33.0% |
| | 55–65 years | 32 | 14.5% |
| | > 60 years | 1 | 0.5% |
| Competence level in palliative care (*n* = 222) | None | 58 | 26.1% |
| | A1 Primary Palliative Care (approx. 24 h) | 23 | 10.4% |
| | A2 Primary Palliative Care (approx. 40 h) | 29 | 13.1% |
| | B1 Primary Palliative Care (approx. 80 h) | 27 | 12.2% |
| | B2 Specialized Palliative Care (approx. 280 h) | 50 | 22.5% |
| | C Highly Specialized Palliative Care (> 1800 h) | 35 | 15.8% |
| Profession (*n* = 246) | Care service manager | 4 | 1.6% |
| | Dietician | 1 | 0.4% |
| | (Family) physician | 32 | 13% |
| | Nurse | 179 | 72.8% |
| | Nursing assistant | 9 | 3.7% |
| | Nursing scientist | 9 | 3.7% |
| | Physician assistant | 3 | 1.2% |
| | Pastor | 6 | 2.4% |
| | Psychologist | 1 | 0.4% |
| | Teacher in nursing care | 2 | 0.8% |
| Work setting (*n* = 206) | Ambulant care | 21 | 10.2% |
| | Doctor's office (as independent) | 4 | 1.9% |
| | Doctor's office (as employee) | 6 | 2.9% |
| | Hospice | 1 | 0.5% |
| | Hospital | 130 | 63.1% |
| | Nursing home | 36 | 17.5% |
| | Psychiatry | 1 | 0.5% |
| | University | 4 | 1.9% |
| Patient groups (*n* = 246) | Adolescents with diseases | 1 | 0.4% |
| | Children with diseases | 4 | 1.6% |
| | Patients in acute care | 13 | 5.3% |
| | Patients in ambulant care | 23 | 9.3% |
| | Patients in nursing homes | 25 | 10.2% |
| | Patients in palliative care and at the end of life | 62 | 25.5% |
| | Patients with cancer | 80 | 32.5% |
| | Patients with chronic diseases | 30 | 12.2% |
| | Patients with dementia | 25 | 10.2% |
| Work experience in years (*n* = 243) | < 1 year | 41 | 16.9% |
| | 1–10 years | 53 | 21.8% |
| | 11–20 years | 65 | 26.7% |
| | 21–30 years | 54 | 22.2% |
| | 31–40 years | 28 | 11.5% |
| | > 40 years | 2 | 0.8% |

**Table 2. Single-word analysis: Associations with food refusal.**

| Category | Sub-categories |
|---|---|
| Physical aspects (31%; *n* = 84) | • 23%; *n* = 62: Loss of appetite; nausea, vomiting<br>• 3.6%; *n* = 10: Pain<br>• 3%; *n* = 8: Weakness<br>• 1.5%; *n* = 4: Dyspnea |
| End-of-life (26.6%; *n* = 72) | • 14%; *n* = 38: Dying<br>• 5.5%; *n* = 15: End-of-life<br>• 3.7%; *n* = 10: Terminal<br>• 2.6%; *n* = 7: Wish to die<br>• 0.7%; *n* = 2: Fasting to death |
| Ethical aspects (19.1%; *n* = 52) | • 8.5%; *n* = 23: Autonomy<br>• 7%; *n* = 19: Self-determination<br>• 3.6%; *n* = 10: Attitude |
| Mental-cognitive aspects (18.1%; *n* = 49) | • 6.3%; *n* = 17: Fear<br>• 5.5%; *n* = 15: Dementia<br>• 4.1%; *n* = 11: Withdrawal, disgust<br>• 2.2%; *n* = 6: Desperation, anger, resignation |
| Modes of reaction and behavior (5.2%; *n* = 14) | • 3%; n = 8: Health professionals<br>• 2.2%; *n* = 6: Patients |

Word frequency analysis revealed that participants primarily associate *loss of appetite*, *nausea*, and *vomiting* and secondarily associate *dying* with food refusal and with the metaphor of *closed lips*.

The code relations analysis revealed that the category *physical aspects* is closely related to the sub-codes *dying* and *fear* (see Table 3).

Additionally, the category *mental-cognitive aspects* is closely connected with *dying*. The chi-square test revealed that the lines and columns in Table 3 are interdependent ($p < 0.0001$; CI = 0.05). To use chi-square test based on chi-square approximation, theoretical frequencies may not be smaller than five. Therefore, the category *modes of reactions and behavior* was not taken into account in testing, as the absolute code association amounted to just four codes (Table 4).

The test of significance per cell revealed which associations participants had with the refusal to eat. While *physical aspects* are significantly associated with *dying* and *fear*, there are no significant relations to loss of appetite and other categories. Concerning the associations with food refusal, the following deduction is possible and can be visualized in Fig 1:

In view of Fig 1, it is clear that the identified associations with the slogan *Food refusal–and the lips remain closed*! do not correspond to the desired picture, which is promoted by the palliative care scene.

Using open questions revealed that many issues in the context of food refusal remained unanswered. The spectrum reaches from reasons over interpretations, possibilities for action, consequences, and ethical issues to limits concerning team resources, self-determination and

**Table 3. Code relations based on single-word analysis.**

| Code Relations/Associations | Loss of appetite | Dying | Fear | Dementia | Self-determination | Autonomy | Total |
|---|---|---|---|---|---|---|---|
| Physical aspects | 26 17.9% | 58 40% | 40 27.6% | 21 14.5% | - | - | 145 36.8% |
| End-of-life | 14 14.4% | 11 11.3% | 22 22.7% | 20 20.6% | 14 14.4% | 16 16.5% | 97 24.6% |
| Ethical aspects | - | 23 67.6% | - | 11 32.4% | - | - | 34 8.6% |
| Mental-cognitive aspects | 18 15.7% | 34 29.6% | 20 17.4% | 18 15.6% | 12 10.4% | 13 11.3% | 115 29.2% |
| Modes of reaction and behavior | - | 3 100% | - | - | - | - | 3 0.8% |
| Total | 58 | 129 | 82 | 70 | 26 | 29 | 394 |

**Table 4. Testing the significance levels of code relations according to Fisher's exact test.**

|  | Loss of appetite | Dying | Fear | Dementia | Self-determination | Autonomy |
|---|---|---|---|---|---|---|
| Physical aspects | 0.189 | 0.014* | 0.015* | 0.219 | <0.001** | <0.001** |
| End-of-life | 1.000 | <0.001** | 0.667 | 0.446 | <0.001** | <0.001** |
| Ethical aspects | 0.005** | <0.001** | <0.001** | 0.033* | 0.150 | 0.159 |
| Mental-cognitive aspects | 0.757 | 0.553 | 0.279 | 0.563 | 0.073 | 0.088 |

**p<0.01

*p<0.05

professional responsibility. The ethical question of whether food refusal can be regarded as a natural death, suicide or starvation is highly important.

## Experiences with food refusal

In total, the participants made 62 statements concerning the frequency of food refusal. These statements included: the phenomenon is frequently occurring in practice (up to 45.9%), it is experienced more frequently (up to 16.4%), it is experienced rarely (up to 29.5%), and it has not yet occurred (up to 8.2%). Participants reported that the phenomenon of food refusal mainly concerns the long-time care setting and is rarely seen in outpatient or hospice care.

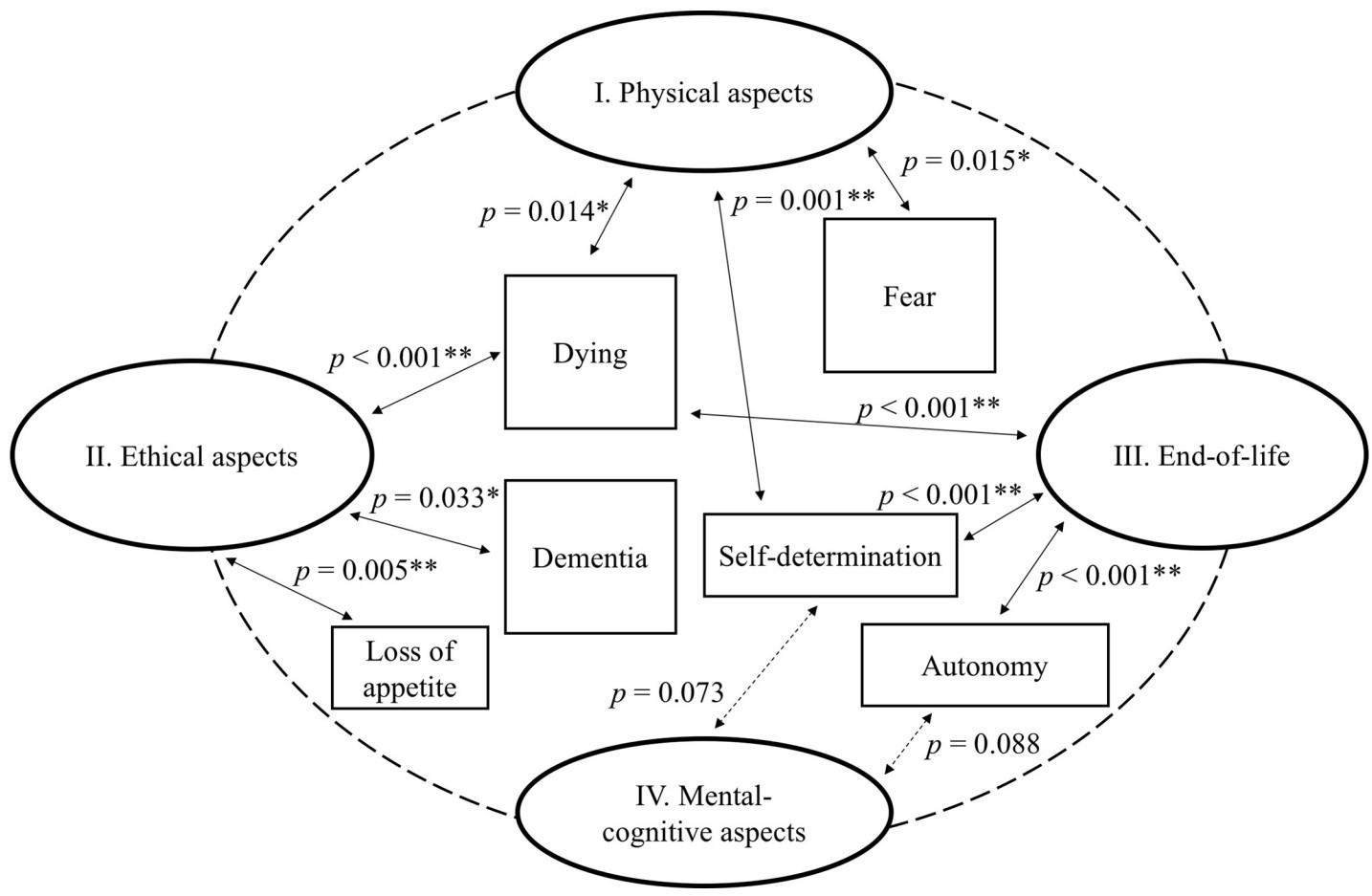

**Fig 1. Main associations and subcode relations.**

**Reasons for food refusal in transitional and terminal stages.** According to the participants, the main reason for food refusal is *no longer wanting* food. This is a characteristic of unnecessary and additional suffering as a cause for accelerated dying and the induction of premature death. The weakening of the body is deliberately supported. The lost meaning of eating and drinking is an additional reason for food refusal. When feelings of hunger and thirst are reduced at the end of life, social pressure about food intake is regarded as a burden.

> Nurse: "*Many patients can't eat any more at all due to their primary illness. Particularly in older people, I observed that they refuse to eat–in the hope to die sooner.*" (Participant: 168)

Additionally, refusing to eat and drink corresponds to the need for autonomy at the end of life. Fear of dependency and increasing need for care has an effect on the need for autonomy. Food refusal can be considered as the patient's last opportunity to express his will:

> Doctor: "*In an institution, eating is like the last bastion where I can be my own master. Washing, dressing, and mobilizing other persons have taken over for me.*" (Participant: 81)

Participants referred to the impossibility to speak about the phenomenon of food refusal. They were not personally confronted with a situation in which a patient pressed his lips together, but they emphasized that, in some respects, the issue of food refusal is deliberately not discussed. This predominantly refers to reasons leading to food refusal, and less to the fact of not eating anymore. The participants described the wish to die with words like *having enough from life* and *not wanting any more*. This is characteristic of the meaninglessness of eating at the end of life when the body, due to disease and proximity to death, no longer signals a demand. In this sense, food refusal is interpreted as a retreat from life.

Transition to the terminal stage at the end of life is a natural process. Patients themselves intuitively perceive the beginning of another stage:

> Nurse: "*For old people (with dementia), this is often the only way to show that they have reached the end of their life. It is a signal that they are withdrawing into themselves. They accept the weakness of their body. From my point of view, they are not suicidal. They behave authentically, according to their perceived reality. Insofar as they receive respect, they are quiet and relaxed. In my view, the final stage of a life can't be defined primarily by means of medical diagnoses or years of life. It is felt intuitively and therefore difficult to communicate and to be taken seriously.*"(Participant: 90)

The end of life cannot accurately be predicted, but reduced eating and drinking habits already indicate the terminal stage. While food intake is reduced, patients continue to express their wish to drink for a long time. Experience shows that the wish for food refusal emerges simultaneously with the wish to die. While the reduction of eating and drinking is rather implicit at the beginning, food refusal at the end of life is explicit and deliberate:

> Nurse: "*In my experience, these statements often express the wish to end one's life without a long time of suffering. I have observed that food refusal was no longer an issue after the patients had been informed about the possibilities of palliative care. They knew that we will do our best to reduce distressing symptoms. In the terminal stage, eating and drinking are recurrent issues. There may be some days or weeks before one can observe a reduced amount of eating and drinking. The body seems to not need this anymore. Eating and drinking lose their significance or receive another significance.*"(Participant: 91)

According to the participants, the deliberate decision to renounce eating and drinking can be characterized as the patient's last opportunity to express his will. Refusing to eat means finally being allowed to die. From the participants' point of view, the decision for *voluntary stopping of eating and drinking* is taken before the transition to the terminal stage. Thereby, terminality is prematurely induced. A characteristic of *fasting to death* is the clearly expressed and resolute will to maintain fasting until the end.

From the perspective of the people themselves, eating and drinking can assume a compulsory dimension and exert pressure leading to exasperation. In this context, the participants mentioned the metaphor of *a satisfied life*. Deliberate and voluntary stopping of eating and drinking is the expression of the patient's strong personality. It can be characterized as the last remaining chance to exert autonomy and power in a situation of dependency. Food refusal can result in a *fight against the others*. This is interpreted in various ways by professionals and relatives. Food refusal is partly regarded as an act of defiance against relatives, nurses and doctors, and partly in the context of self-determination and responsibility for oneself. It is considered as an expression of the patient's will or is labeled as a form of challenging behavior. It becomes clear that food intake and drinking imply a balancing act between burden and relief for the affected person, resulting in situations affording highly sensitive ethical decision-making for all persons involved.

Food refusal is associated with not only *voluntary stopping of eating and drinking* but also the issue of suicide:

> Nurse: *"[Food refusal] is rarely accepted by health professionals. Frequently it is put on the same level with suicide. There is a high potential of conflict between relatives, staff and patients."* (Participant: 20).

Participants issued a warning not to interpret closed lips generally as an expression of voluntary stopping of eating and drinking. However, food refusal represents an active component with the aim of accelerated dying. Participants regarded it as a "real" alternative to assisted suicide and palliative sedation.

**Consequences of food refusal.**   Reduced food intake and drinking causes various reactions and conflicts because it is difficult to accept for doctors, nurses, other health professionals and clergies. They feel personally affected and discuss this issue with the team. Doctors have less respect for the patient's decision to refuse food.

> *"(. . .) veins are punctured, and if nothing else works, a central venous catheter, a gastric tube or a PEG [percutaneous endoscopic gastrostomy] are applied. In my experience, doctors rarely respect a patient's refusal to eat, and I wonder where self-determination comes into effect if patients simply refuse food and it is administered without their consent, supposing to know what is best for the patient. I think the issue of voluntary stopping of eating and drinking causes much insecurity in health professionals, particularly in doctors. But nurses also intervene, assuming that they have done something good. You can't let somebody starve to death–this sentence I have heard again and again in ethical discussions with my colleagues. In my opinion, it is really possible to do this. If the person has reflected it and has decided this for himself, then this seems to me the most natural way to die."* (Participant: 70).

Many questions concerning food refusal remain unanswered (Table 3) and cause uncertainty due to various consequences. For example, oral medication is no longer possible, frequently resulting in the application of artificial nutrition. This is justified by arguments referring to *the necessity* and *the compulsion* to act in this way. Additionally, the participants

mentioned that persons with loss of appetite at the end of life wished to receive parenteral nutrition:

> *"Oncological conditions are often associated with loss of appetite. I make a difference between this and real food refusal. Persons with loss of appetite partially wish to receive parenteral nutrition. When I worked in long-time care I experienced food refusal more frequently than in patients with an oncological condition."* (Participant: 118)

If parenteral interventions are no longer possible, infusion therapy represents the last remedy. Health professionals' fear and stress are often associated with their difficulty of accepting food refusal, resulting in various forms of forcing the patient to eat. Participants also mentioned that patients revised their decision for food refusal, as they experienced too-intense feelings of hunger. They accepted high-calorie drinks.

The only positive association mentioned by the participants was related to less *death rattle*, less edema and less ascites in the terminal stage due to restricted drinking. The participants assume that food refusal can be performed in a *calm* and *detached* way if the consequences are taken seriously and treated with respect.

**Challenges concerning relatives.**   Responding to relatives' reactions towards food refusal is challenging for the participants. There is a high potential for conflict between relatives, professionals and the patient with regard to food refusal. Respecting and tolerating the decision is described as demanding. From the relatives' point of view, difficulties can arise particularly because of their worries and their lack of understanding:

> *"For relatives, it is often difficult to understand because eating is vitally important. If someone doesn't eat, he will die. And if someone actively decides not to eat, the reason for his death is not the illness, but the patient himself causes his death. There are often misunderstandings in daily professional practice. A strong sense of tact and phantasy are needed to do justice to the patients. Phantasy in order to possibly stimulate the appetite again and to ensure quality of life for the patient or sense of tact in order to respect the patient's wish and to communicate it to his relatives and to health professionals of other disciplines."* (Participant: 89)

Primarily, relatives associate food refusal with *starving to death* and *dying from thirst*, which is terrifying for them. Relatives turn to professionals with pressing questions: *You can't let him starve to death, can you*? or *You don't let him die him of hunger, do you*?. To signify the decision against life and in favor of death by means of food refusal is difficult for relatives to understand. Respecting this decision is described as particularly challenging for them. It is the task of professionals to mediate between patients and relatives. The patients' situation and the contact with relatives causes feelings of helplessness and powerlessness in health professionals.

Uncertainty associated with food refusal evokes fear. Relatives often demand that professionals spare no effort to ensure food intake and drinking. They insist that the patient should eat at any rate by offering the favorite meal or asking for parenteral nutrition. Additionally, relatives often imagine that the patient will gather their strength again. They try to influence professionals as well as the patient by insisting, persuading and convincing.

## Attitudes and reactions towards food refusal

**Attitudes in the context of patient autonomy.**   Attitudes and personal convictions concerning food refusal are very heterogeneous, depending on the participants' personal knowledge and professional networks in practice:

*"It is accepted very differently by nurses. They respond variously. Depending on their level of knowledge or further education, it generates discussions."* (Participant: 37)

The participants' attitude towards food refusal was expressed in terms of *respect* (52%; *n* = 42 nominations) and *acceptance* (48%; *n* = 38 nominations). While *respect* means to take the decision into account, the term *acceptance* implies a positive evaluation of the decision. Half of the participants either respected or accepted the decision. The attitudes of principally respecting closed lips, the personal will, the decision as well as the patients' self-determination were not shared by all participants. Even if a person suffers from dementia, the decision has to be taken seriously. *'No' means 'no' and remains 'no'*, as a participant said. If persons cannot articulate themselves, their decision at least has to be accepted. *Acceptance* implies permitting natural consequences. With regard to food refusal, an open and accepting attitude is a precondition for adequately responding to this phenomenon. The autonomy of every person to follow this deliberately chosen way has to be taken seriously. Food refusal allows expressing one's will in a self-determined, autonomous way, even under conditions of cognitive limitations. According to the participants, food refusal represents a basic right of the affected persons. The wish and will of the patient have the highest priority.

The participants' attitude is constituted within a field of tension between their personal interpretation of the metaphor *closed lips*, the attribution of eating and drinking and the acceptance of the natural end of life. From their point of view, *starving to death* and *dying of thirst* are only given when the patient really suffers from hunger and thirst. Thus, the term *refusal* is not acceptable, as it reduces a complex issue in a disadvantageous way. It is not a question of maintaining life by means of eating and drinking but of ensuring quality of life based on reduced food intake. The term *refusal* tends to hide aspects concerning quality of life.

**Challenges of responding to food refusal.** Several challenges are characteristic for responding to food refusal. On the one hand, decisions have to be made; on the other hand, much phantasy is afforded to maintain patients' quality of life.

Oscillating between practical and ethical decision-making represents another challenge. On the one hand, food is overrated in the final stage of life; on the other hand, a sense of tact and phantasy are necessary to do justice to the patients. It is important to know the patient well and to establish trust. In particular, knowing the patients' needs is essential for accepting food refusal. If trust is missing, the decision is accepted only after clarification or informed consent. Biographical work in the context of food refusal is essential to ensure traceability:

*"I frequently care for patients who wanted to make use of food refusal in order to end their lives. However, this mostly remained a remark, and I am very pleased about that. For me, it is hard to imagine that someone who has reached the end of his life but is physically still relatively fit wants to end his life in this way. I assume that starving to death is quite painful. In my experience, this remark often suggests that patients wish to end their life without a long time of suffering. Therefore, I found that food refusal was no longer an issue after patients were informed about the possibilities of palliative care."* (Participant: 89)

Personally, food refusal is understandable, but enduring it and holding on are challenging. Palliative care allows the caregiver to respond to food refusal in a professional way. The team attitude is highly important. Interdisciplinary communication with regard to food refusal is mentioned as an additional challenge. In critical situations, a firm basis of communicative skills may prove effective.

**Searching for reasons and "doing it differently".** Responding to food refusal in a professional way implies searching for reasons and understanding them. Besides physical symptoms,

psychological and social dimensions are also mentioned. Additionally, symptom clusters or side effects of medications or chemotherapies are described as possible reasons for food refusal. Depending on individual patient situations, food refusal should be regularly assessed. Besides treatable reasons, it is also necessary to assess the patients' readiness for food refusal.

The professional response to food refusal means "doing it differently" and to offer alternatives. Serving meals is time-consuming. According to the participants, cheering up is not the same as persuading. The fine line between cheering up and persuading should not be passed. Thus, it is important to regularly assess the situation, without asking explicitly at every meal. Additionally, it should be assessed to what extent another person offers food and drink. An additional aspect of the *fine line* consists of preventing *over-protection* of persons at the end of life, as a participant said. Referring to *compulsion* or *necessity* does not affect the basic problem resulting in refusal and renouncement. Meal times are often too restrictive and should be adjusted to the needs of the affected person. Despite refusal, it is indispensable to offer meals and to plan sufficient time for this. Another element to be considered is communication as a form of caring. If eating is no longer possible, it is necessary to offer conversations to ensure human attention until death. *Doing it differently* means to use phantasy to stimulate the patient's appetite. Finally, participation in rather than quantity of social life is important.

A professional response requires introducing educational interventions, as time for conversations and emotional support is often missing. Information and instructions are very important in understanding the process leading to food refusal:

*"(. . .) it is important to communicate this issue and to know the attitude of the patient. Relatives need information and counselling in order to do something good for the patient, for example helping with oral care. For relatives, it is often very difficult to endure."* (Participant: 22)

In the case of food refusal, discussions with relatives are inevitable. They need information about the reasons and consequences of this phenomenon. Informative material for relatives is often missing. Doctors and nurses rarely know how relatives experience food refusal and which conflicts can arise within families. Joint decision-making reduces the psychological pressure to consider the positive aspects of food refusal. Finally, it is important to build a solid basis for joint decision-making.

## Discussion

More than half (62.3%) of the professionals frequently or more frequently experience that patients refuse to eat. Taking into account the difficulties of recognizing food refusal, especially when patients do not articulate themselves, we assume these frequencies could even increase [3]. In examining the reasons for food refusal, it becomes clear that the patients have completed their life and are not willing to live anymore. Perhaps the person is already in the process of dying, where the reduction of food is common [7], and the food refusal is only an expression of that. However, to be sure of that, depression or other factors should be excluded. Another reason is the patient's fear of losing their autonomy. This can be explained by the fact that a patient's autonomy can be compressed by external influences, such as pain or fatigue [45]. As a result, patients would rather end their lives than to enter into a dependency relationship [46]. A special challenge for professionals arises when patients cannot or will not articulate themselves. If the patient suffers from dementia, the patient's behavior must be interpreted by the professionals. Since the reasons for food refusal can be diverse, the situation is difficult to assess [5].

One of the consequences of food refusal is discussion. The refusal causes embarrassment and leads to different reactions among the team. In addition to the different attitudes of the professionals, the communication is made difficult due to insufficient knowledge among the professionals regarding possible interventions and the reasons and causes for food refusal. To promote good inter-professional communication, an educational program like COMFORT can serve as support [47].

The results regarding the attitudes and reactions of the professionals show that the autonomy of the patient is the highest good and is to be respected. Even in the case of patients with limited decision-making capacity, the refusal to eat is acceptable. However, it must first be ensured that the food refusal does not take place due to external influences (such as pain or food has no taste). Clarification of the cause can take a lot of time and discussion. However, the patient must not be persuaded or even forced to eat.

## Strengths and limitations

The strength of this study can be seen in the transformation process of qualitative data into quantitative data to synthesize the association into a model which, in addition to the main associations, also calculates the proximity to subcodes. The limitation of this study is the main reliance on a method that is still little used and therefore difficult to comprehend. For this reason, care was taken to describe the procedure as precisely as possible. Due to limited resources, we were not able to make subgroup comparisons of response behavior, for example, to age, although the findings would certainly be interesting. A secondary data analysis in this respect is not excluded. The low response rate of 17.5% shows that the issue of food refusal is given little importance and relevance in the care of palliative care patients. However, the results clearly show that a discussion in the form of further training is necessary to achieve a rethinking of the associations around food. The results are to be regarded as preliminary and should be validated by further research.

## Implications for practice

The results of the keyword associations show that the term *food refusal* is primarily associated with physical and ethical aspects and with the end-of-life. Considering the relationships of the sub-codes, mainly negative aspects are significantly linked, such as dying, dementia and fear. Positive aspects include self-determination and autonomy. The palliative care scene should deal with which image should be based on food refusal. Furthermore, what measures are necessary to change the existing associations? Our recommendation is to provide targeted training based on the SENS model [48] to improve patient care. The SENS model covers the objectives that the Swiss Federal Office of Public Health has called for in the National Palliative Care Strategy [49] and is therefore suitable for implementation and provides a structural basis for palliative care in Switzerland. The SENS model is based on the gold standard framework from England but has been more strictly oriented on the patient perspective for use in Switzerland. The four goals can be defined as the four *S*s of palliative care: self-efficacy in symptom management; sense of coherence in decision making; security; and support of the family, including bereavement [48]. The four main associations are to be designed as follows (see Fig 2).

First association: Food refusal is a natural process in the dying phase [50]. For this reason, we want to link the focus of the physical aspects less with fear and dying. Rather, the patients should be taught how to organize their own symptom management. Second association: We have connected the end-of-life and ethical aspects together by proximity in the subgroups. The patient is to make self-determination decisions, which should be respected [48]. It should also be borne in mind that, in the event of a limited decision-making competence of the patient (such as patients

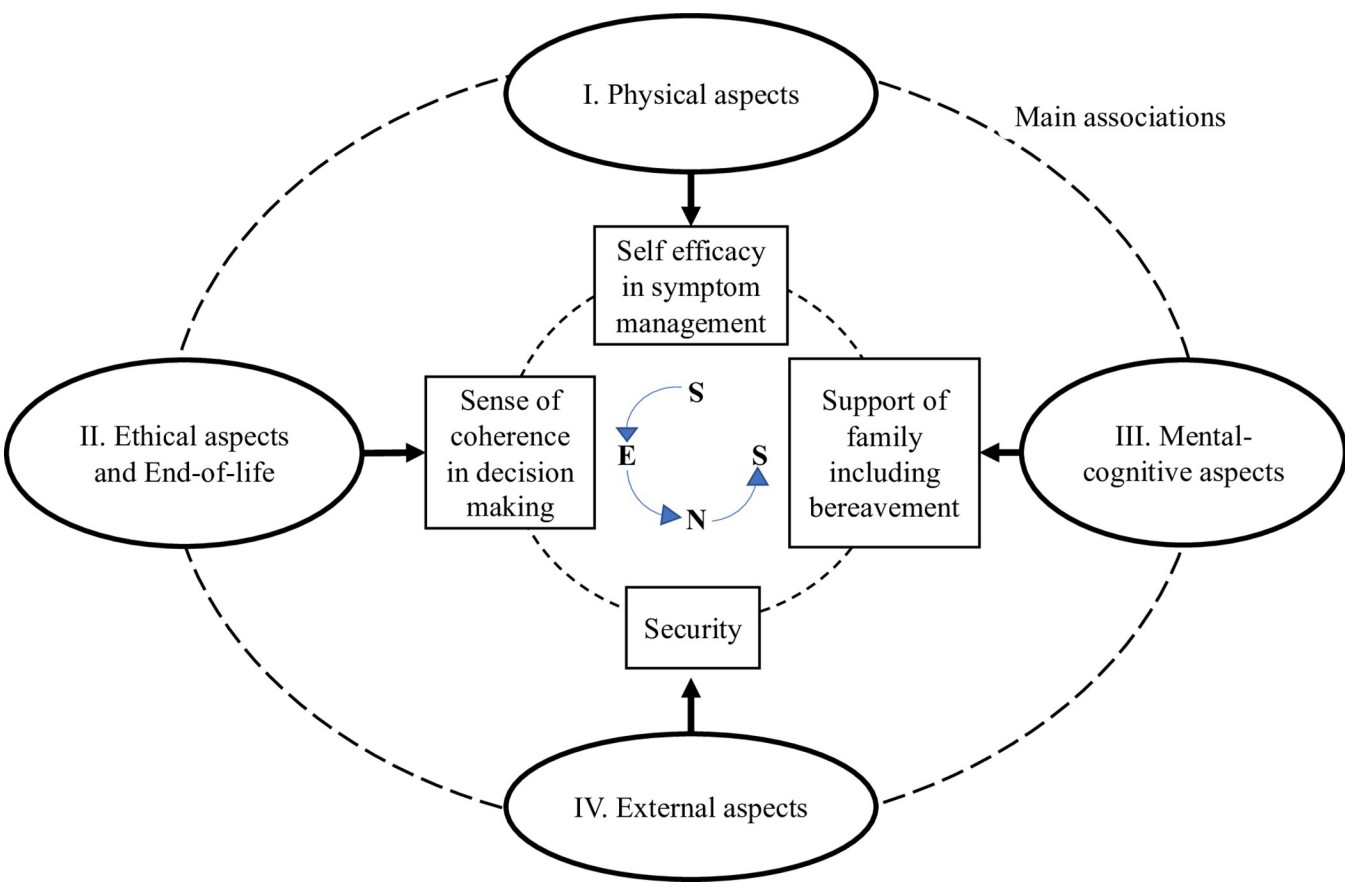

**Fig 2. Target associations on the topic of food refusal using the SENS model [48].**

with advanced dementia), the decision is made by the relatives or the legal guardian. However, food refusal is met with resistance. The thought of someone starving triggers negative feelings, particularly in relatives [51]. This leads us to the third association. Relatives need support during the process and beyond death. Finally, in the fourth association, we added the external aspect of security. A network must be ensured in outpatient and inpatient care.

## Conclusions

Patients who refuse food force relatives and professionals to deal with a great challenge. The clarification of food refusal requires extensive knowledge, a lot of time and clarifying discussions. While the associations of food refusal have thus far been negatively connoted, a need for information, in the sense of further training, can be derived. It is also advisable to examine the affected persons in a differential diagnosis and to involve palliative care experts via conciliatory services. This can reduce anxieties and build a better communication culture.

   This study has shown the various aspects associated with food refusal. The importance of inter-professional co-operation, the discussion with the patient and the involvement of relatives are highlighted. On the one hand, it is important to clarify the cause of food refusal; in any case, care should be taken not to put pressure on the patients. On the other hand, we need to consider our hidden preconceptions in regard to daily-used keywords in practice to prevent prematurely mislabeling patients' attitudes. When in doubt, the patient's will should always be respected.

We recommend that the topic of food refusal be prepared by means of concept analyses and to promote observation studies to achieve a deeper understanding.

## Acknowledgments

Prof Dr. Wilfried Schnepp passed away before the submission of the final version of this manuscript. Prof. Dr. André Fringer accepts responsibility for the integrity and validity of the data collected and analyzed.

## Author Contributions

**Conceptualization:** André Fringer, Stefan Ch. Ott.

**Data curation:** André Fringer, Sabrina Stängle, Daniel Büche.

**Formal analysis:** André Fringer, Sabrina Stängle, Stefan Ch. Ott.

**Funding acquisition:** André Fringer.

**Investigation:** Daniel Büche.

**Methodology:** André Fringer, Sabrina Stängle, Daniel Büche, Stefan Ch. Ott, Wilfried Schnepp.

**Project administration:** André Fringer.

**Resources:** Sabrina Stängle, Daniel Büche, Wilfried Schnepp.

**Software:** Sabrina Stängle, Stefan Ch. Ott.

**Supervision:** Wilfried Schnepp.

**Validation:** André Fringer, Stefan Ch. Ott.

**Visualization:** André Fringer, Sabrina Stängle, Stefan Ch. Ott.

**Writing – original draft:** André Fringer, Sabrina Stängle.

**Writing – review & editing:** André Fringer, Sabrina Stängle, Daniel Büche, Stefan Ch. Ott, Wilfried Schnepp.

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
