## [Decision Letter · Decision Letter 0]

11 Dec 2019

PONE-D-19-21281

Professionals´associations on food refusal: Data transformation qual -> quan

PLOS ONE

Dear Mrs. Stängle,

Thank you for submitting your manuscript to PLOS ONE. After careful consideration, we feel that it has merit but does not fully meet PLOS ONE’s publication criteria as it currently stands. Therefore, we invite you to submit a revised version of the manuscript that addresses the points raised during the review process.

Please adress the minor revisions sugested by both reviewers.

We would appreciate receiving your revised manuscript by Jan 25 2020 11:59PM. To enhance the reproducibility of your results, we recommend that if applicable you deposit your laboratory protocols in protocols.io, where a protocol can be assigned its own identifier (DOI) such that it can be cited independently in the future. For instructions see: http://journals.plos.org/plosone/s/submission-guidelines#loc-laboratory-protocols

We look forward to receiving your revised manuscript.

Kind regards,

Manuel Fernández-Alcántara, Ph.D.

Academic Editor

PLOS ONE

Journal Requirements:

1. Please correct your reference to "p=0.000" to "p<0.001" or as similarly appropriate, as p values cannot equal zero.

2. Please remove any abbreviations and symbols from your title in your manuscript and on the online submission form

4. Please ensure authors names are not duplicated on the online submission form

Reviewers' comments:

Reviewer's Responses to Questions

**Comments to the Author**

1. Is the manuscript technically sound, and do the data support the conclusions?

Reviewer #1: Yes

Reviewer #2: Yes

2. Has the statistical analysis been performed appropriately and rigorously? 

Reviewer #1: Yes

Reviewer #2: Yes

3. Have the authors made all data underlying the findings in their manuscript fully available?

Reviewer #1: Yes

Reviewer #2: Yes

4. Is the manuscript presented in an intelligible fashion and written in standard English?

Reviewer #1: Yes

Reviewer #2: Yes

5. Review Comments to the Author

Reviewer #1: Only 17.5% of participants completed the study. This percentage may indicate the little consideration of a topic such as the refusal of food. This aspect should be well emphasized and discussed.

It should be noted that the practical application of the results of this study must be confirmed by other studies, before being considered valid for influencing the guidelines.

Reviewer #2: It is an interesting and good article.

The manuscript focuses on interesting and sensitive issue that has important ethical implications for medical practice.

It is adequately written.

According the authors, the strength and the limitation of the study are methodological aspects. Authors say: “The limitation of the study is mainly due to a method that is still little used and therefore difficult to comprehend. For this reason, care was taken to describe the procedure as precisely as possible”. However, the exactly name of method used is omitted. I consider it would be important to justify the use of mixed methods and briefly explain their main characteristics. As the authors indicate, if the method used is difficult to comprehend, this would help to understand it. The type of design used is also omitted. Indicating, explaining and justifying it would also help to understand the methodological questions of the study. In this way, limitation of the study could be minor.

In relation to these same methodological questions and the difficulty for their understanding (as it is indicated by the authors themselves), it would be necessary to explain what the colors of the cells in table 3 mean. In addition, it would be more advisable not to use colors as indicators of specific issues. (There may be many readers who prefer to print the article to read it and a color printer is not available for them.)

In the same way, table 4 could be ameliorated. It would be more appropriate to use asterisks (*) to indicate statistically significant differences (not the color red). Further, why the meaning of the color red is explained but not the meaning of the bold font, gray color or square brackets []?

It is possible to clarify why lines 195-196 indicate that the category end-of-life was not clearly associated with dying? (p <0.001)

It would be necessary to indicate in the Discussion Section future lines of research that extend the objective of this study, for example: are there differences between the answers of the participants depending on their profession or their work experience in years?

Other minor aspects:

The full title is not precise / accurate. It would better a title that specifies more the content of the manuscript. (It is possible because the journal allows more characters to be used in the title). Further, it would be advisable that full title and short title not be exactly the same.

It is advised to improve the Introduction section. It is too short. The state of the art is scarcely presented.

The idea transmitted on lines 180 and 181 is repeated on lines 184 and 185.

After the title of table 4 appears: “Significance per cell”. This must be removed. They are words that are isolated. In addition, this is already indicated at the beginning of line 192.

References should be reviewed. The Vancouver style (style demanded by Plus One) has not been adequately used.

In table 4 “ : ” must be eliminated after Physical aspect.

6. PLOS authors have the option to publish the peer review history of their article (what does this mean?). If published, this will include your full peer review and any attached files.

Reviewer #1: Yes: Marcellino Monda

Reviewer #2: No

---

## [Author Response · Author response to Decision Letter 0]

17 Jan 2020

Many thanks for your hints and constructive criticism. We have revised all the points you have made. Please use the document "Response to Reviewer" in which all points were answered one below the other.

---

## [Editor Report · Decision Letter 1]

27 Jan 2020

PONE-D-19-21281R1

The associations of palliative care experts regarding food refusal: a cross-sectional study with an open question evaluated by triangulation analysis

PLOS ONE

Dear Mrs. Stängle,

Thank you for submitting your manuscript to PLOS ONE. After careful consideration, we feel that it has merit but does not fully meet PLOS ONE’s publication criteria as it currently stands. Therefore, we invite you to submit a revised version of the manuscript that addresses the points raised during the review process.

You have addressed the main concerns of the reviewers and the manuscript has remarkably improved. There is just a minor point that should be considered before final acceptance. Authors have made modifications in the conclusion of the research (in the main file) and this modifications should be also included in the abstract section. Please modify the abstract and re-submit the manuscript.

We would appreciate receiving your revised manuscript by Mar 12 2020 11:59PM. To enhance the reproducibility of your results, we recommend that if applicable you deposit your laboratory protocols in protocols.io, where a protocol can be assigned its own identifier (DOI) such that it can be cited independently in the future. For instructions see: http://journals.plos.org/plosone/s/submission-guidelines#loc-laboratory-protocols

We look forward to receiving your revised manuscript.

Kind regards,

Manuel Fernández-Alcántara, Ph.D.

Academic Editor

PLOS ONE

---

## [Author Response · Author response to Decision Letter 1]

28 Feb 2020

I have revised the abstract of the conclusion and uploaded the new document. During the overview I also noticed that an image was missing (Fig2), which I also uploaded.

Thank you very much and kind regards,

Sabrina

---

## [Editor Report · Decision Letter 2]

23 Mar 2020

The associations of palliative care experts regarding food refusal: a cross-sectional study with an open question evaluated by triangulation analysis

PONE-D-19-21281R2

Dear Dr. Stängle,

We are pleased to inform you that your manuscript has been judged scientifically suitable for publication and will be formally accepted for publication once it complies with all outstanding technical requirements.

With kind regards,

Manuel Fernández-Alcántara, Ph.D.

Academic Editor

PLOS ONE
---

## [Editor Report · Acceptance letter]

25 Mar 2020

PONE-D-19-21281R2 

The associations of palliative care experts regarding food refusal: a cross-sectional study with an open question evaluated by triangulation analysis 

Dear Dr. Stängle:

I am pleased to inform you that your manuscript has been deemed suitable for publication in PLOS ONE. Congratulations! Your manuscript is now with our production department. 

With kind regards,

on behalf of

Dr. Manuel Fernández-Alcántara 

Academic Editor

PLOS ONE